# miRNAs and lncRNAs as Novel Therapeutic Targets to Improve Cancer Immunotherapy

**DOI:** 10.3390/cancers13071587

**Published:** 2021-03-30

**Authors:** Maria Teresa Di Martino, Caterina Riillo, Francesca Scionti, Katia Grillone, Nicoletta Polerà, Daniele Caracciolo, Mariamena Arbitrio, Pierosandro Tagliaferri, Pierfrancesco Tassone

**Affiliations:** 1Department of Clinical and Experimental Medicine, Magna Graecia University of Catanzaro, 88100 Catanzaro, Italy; caterina.riillo1@studenti.unicz.it (C.R.); k.grillone@unicz.it (K.G.); nicoletta.polera@studenti.unicz.it (N.P.); daniele.caracciolo1@studenti.unicz.it (D.C.); tagliaferri@unicz.it (P.T.); tassone@unicz.it (P.T.); 2Institute of Research and Biomedical Innovation (IRIB), Italian National Council (CNR), 98164 Messina, Italy; scionti@unicz.it; 3Institute of Research and Biomedical Innovation (IRIB), Italian National Council (CNR), 88100 Catanzaro, Italy; mariamena.arbitrio@cnr.it

**Keywords:** noncoding RNA, microRNA, miRNA, long noncoding RNA, lncRNA, RNA therapeutics, immunotherapy, cancer

## Abstract

**Simple Summary:**

Cancer onset and progression are promoted by high deregulation of the immune system. Recently, major advances in molecular and clinical cancer immunology have been achieved, offering new agents for the treatment of common tumors, often with astonishing benefits in terms of prolonged survival and even cure. Unfortunately, most tumors are still resistant to current immune therapy approaches, and basic knowledge of the resistance mechanisms is eagerly awaited. We focused our attention on noncoding RNAs, a class of RNA that regulates many biological processes by targeting selectively crucial molecular pathways and that, recently, had their role in cancer cell immune escape and modulation of the tumor microenvironment identified, suggesting their function as promising immunotherapeutic targets. In this scenario, we point out that noncoding RNAs are progressively emerging as immunoregulators, and we depict the current information on the complex network involving the immune system and noncoding RNAs and the promising therapeutic options under investigation. Novel opportunities are emerging from noncoding-RNAs for the treatment of immune-refractory tumors.

**Abstract:**

Immunotherapy is presently one of the most promising areas of investigation and development for the treatment of cancer. While immune checkpoint-blocking monoclonal antibodies and chimeric antigen receptor (CAR) T-cell-based therapy have recently provided in some cases valuable therapeutic options, the goal of cure has not yet been achieved for most malignancies and more efforts are urgently needed. Noncoding RNAs (ncRNA), including microRNAs (miRNAs) and long noncoding RNAs (lncRNAs), regulate several biological processes via selective targeting of crucial molecular signaling pathways. Recently, the key roles of miRNA and lncRNAs as regulators of the immune-response in cancer have progressively emerged, since they may act (i) by shaping the intrinsic tumor cell and microenvironment (TME) properties; (ii) by regulating angiogenesis, immune-escape, epithelial-to-mesenchymal transition, invasion, and drug resistance; and (iii) by acting as potential biomarkers for prognostic assessment and prediction of response to immunotherapy. In this review, we provide an overview on the role of ncRNAs in modulating the immune response and the TME. We discuss the potential use of ncRNAs as potential biomarkers or as targets for development or clinical translation of new therapeutics. Finally, we discuss the potential combinatory approaches based on ncRNA targeting agents and tumor immune-checkpoint inhibitor antibodies or CAR-T for the experimental treatment of human cancer.

## 1. Cancer Immunotherapy

The immune system plays a multifaceted role in cancer, promoting eradication or stimulation of malignant cells, in a mutual influenced and complex interaction between tumor and its microenvironment, named cancer immunoediting [1]. This dynamic but not unique and nonlinear mechanism that occurs during cancer onset, progression, and development of drug resistance, includes three phases: (i) elimination, in which both innate and adaptive immune cells, organized in the immunosurveillance network, kill malignant cells through the recognition of tumor-specific molecules expressed on the cell membrane. These molecules, named tumor antigens, act as dangerous signals by stimulating immune cells to avoid tumor development. Specifically, in this step, tumor antigens loaded on Major Histocompatibility Complexes (MHCs) are presented on TCD4+ or TCD8+ lymphocytes, resulting in priming, activation, and trafficking of T cells from lymph nodes to tumor sites. Tumor cell killing occurs via MHC-TCR (T Cell Receptor)-specific interactions. The immune-mediated cancer cell killing elicits antigen release that contributes to the formation of a circular process known as cancer immunity cell cycle [2]. The second step of cancer immune editing is known as (ii) equilibrium and represents an intermediate phase in which the immune system cannot eradicate tumor cells that are characterized by a reduced immunogenicity phenotype and maintains a state of dormancy. (iii) Escape is the third phase during which tumor cells under selective immune pressure undergo genetic and epigenetic changes, leading to the acquisition of immune-evasive properties and enabling them to circumvent immune system defense. In this phase of uncontrolled proliferation, cancer cells can develop clinically detectable tumors [1]. Taking into account the high deregulation of immune cell cycle during cancer onset and progression, a potential effective anticancer therapy is to boost host immune defense against tumor cells and to restore the immune system’s balance. Despite the rapid increase in knowledge and the availability of novel tools in cancer immunotherapy in the past few decades, the roots of cancer immunotherapy are deep. In 1891, Coley, currently considered the “father of the immunotherapy”, observed several cancer spontaneous remissions linked to the development of streptococcal infection. According to this evidence, he tried to inject a mixture of inactivated *Streptococcus pyogenes* and *Serratia marcescens* known as “Coley toxins” into the tumor by developing the first immune-based cancer therapy [3]. From this first attempt, several advances in molecular and clinical cancer immunology have been made. In the last years, an increasing number of immune-based drugs and strategies tailored for specific altered steps of cancer immune cell cycle, such as cancer vaccines, adoptive cell therapy, and immune-modulator agents, are still under development or are clinically approved [4]. In the recent clinical experience, cancer immunotherapy has emerged as a revolutionary therapeutic strategy able to achieve durable response in different types of cancer, and a growing body of evidence suggests the crucial role of ncRNAs in regulating cancer immune response.

Here, we report the regulatory role of the most investigated miRNAs and lncRNAs, summarized in Table 1, which are involved in cancer immunoediting, TME modulation, and immunotherapy resistance. We discuss the potential translational value of ncRNA as promising immunotherapeutic targets in a single-agent strategy or in multi-drug therapeutic approaches and their function as predictive biomarkers for anticancer immune-based therapy.

## 2. Noncoding RNA Biogenesis and Function: An Overview

Genome-wide transcriptome analysis indicates that about 98% of the eukaryotic genome is transcribed as ncRNAs while only a small fraction (≈2%) is translated into proteins [5,6]. NcRNAs are a class of functional RNA molecules without protein-coding abilities. They include “house-keeping” RNAs such as ribosomal RNA (rRNA) and transfer RNA (tRNA) as well as regulatory RNAs. Based on transcript length, regulatory RNAs are divided into two groups: small ncRNAs with <200 nucleotides (nt) and lncRNAs, the most abundant class, with >200 nt length [7,8]. In the past, ncRNAs were considered “evolutionary junk,” but growing evidence suggests that this dark matter of the genome regulate several biological processes via selective targeting crucial molecular pathways [9]. MiRNAs, the widely explored group of small ncRNAs, are encoded at various locations as autonomous or clustered transcriptional units [10]. They are transcribed by RNA polymerase II (Pol II) in primary miRNA transcripts (pri-mRNAs) and then converted by the endonuclease DROSHA and its cofactor DGCR8 in pre-miRNA transcripts [11]. Pre-miRNAs are generated in the nucleus from introns through the splicing machinery [12] and are exported by exportin 5 into the cytosol [13], where they are processed by the RNAse III enzyme DICER and its partner binding protein TRBP [14]. The result is the formation of mature miRNA/miRNA* duplexes, which are rapidly unwinded by an argonaute protein (AGO). The passenger strand (miRNA*) is degraded, whereas the guide strand (mature miRNA) binds to AGO and additional proteins [15] to form the microRNA-induced silencing complex (miRISC) [15]. The main function of miRNAs is the repression of gene expression by binding to the 3′-untranslated regions of target mRNAs [16]. Gene silencing can occur through mRNA destabilization or inhibition of translation [17]. However, in addition to the conventional role in posttranscriptional gene regulation, miRNAs can upregulate target translation by recruiting ribonucleoprotein complexes [18]. MiRNAs are also present in body fluids such as blood, plasma, and urine, where they are associated with carriers or incorporated into vesicles and microparticles [19]. Circulating miRNAs act as signaling molecules transferring their cargo between cells or tissues [20]. Compared to miRNAs, lncRNAs can regulate gene expression at multiple levels in the cell. They are transcribed by the RNAP II complex, similar to protein-coding RNAs [21], or by RNAP III and single-polypeptide nuclear RNA polymerase IV (spRNAP IV) [22] at several loci of genome and in different orientations. As a result, lncRNAs can be classified in intergenic (when the ncRNA is localized between two genes), intronic (when the ncRNA is derived from an intron of a second transcript), sense or antisense (when the ncRNA overlaps one or more exons of another transcript on the same or opposite strand, respectively), and bidirectional (when the ncRNA is transcribed from the opposite strand, in the opposite direction) [23]. The lncRNA transcriptional process shares several features with mRNA transcription, including 5′-capping, 3′-polyadenylation, normal and alternative splicing mechanisms [24], RNA editing processes [25], and patterns of transcriptional activation [26]. However, recently, alternative structures that protect lncRNAs from degradation have been identified [27]. Finally, it has been reported that some lncRNAs are generated from the mitochondrial genome and regulated by nuclear-encoded proteins [28]. Several studies focus on molecular functions of lncRNAs, such as RNA processing, nuclear organization, and transcriptional and posttranscriptional modulation of gene expression [29,30]. They act near their own sites of transcription (cis) or at distant genomic or cellular locations (trans). LncRNAs can mediate epigenetic changes by participating in histone modification and DNA methylation and by recruiting chromatin remodeling complexes to specific genomic loci [31]. It has been reported that they are involved in X chromosome inactivation [32] and genomic imprinting [33]. Furthermore, emerging evidence indicates that lncRNAs are involved in many biological processes, including cell differentiation and development [34], embryogenesis [35], organogenesis [36], and immune response [37]. Due to their involvement in the regulation of crucial cellular pathways, ncRNAs are implicated in various diseases including cancer. Their role as tumor biomarkers as well as their oncogenic or tumor suppressive properties have been demonstrated. Since the role of ncRNAs in immune modulation is emerging and making them potential targets in improving the efficacy of immunotherapeutic approaches, here, we focus on the correlation between ncRNAs and the TME.

## 3. In Silico Approaches to Investigate Immune-Related ncRNA

Inflammation is considered one of the enabling hallmarks of cancer, and it has been estimated that more than 20% of cancers are caused by chronic inflammation. In turn, inflammatory mediators, such as cytokines and chemokines, can regulate the behavior of the immune system and are involved in the events underlying immunotherapy. With the discovery of ncRNAs, a further level of control in immunity and inflammatory processes elicit interests in inflammation-related research. A global view of the inflammatory disease-associated ncRNAs will help to characterize their roles in inflammation and will inspire new approaches to disease therapy. There are quite a lot of databases archiving the disease-associated ncRNAs that are not specifically designed for inflammatory disease. Recently, a few software have been released and available data appear to support a functional interaction of ncRNA players and to target immune-related signatures.

Prabahar A. et al. presented an integrated human immune disease-associated miRNAs database, ImmunemiR, through interactome network, to provide a repository for immune-related disease and miRNA association [38]. The aim was to understand the miRNA’s role and their function in regulating the immune system during inflammation, cancer development and progression, and autoimmune disease. A number of 245 immune miRNAs, associated with 92 OMIM (Online Mendelian Inheritance in Man) disease categories, were identified from databases such as HMDD, miR2Disease, and PubMed literature based on Mesh classification of immune system diseases and compiled as an ImmunemiR database. This last provides both text-based annotation and network visualization options of its target genes, protein–protein interactions, and its disease associations (freely available at http://www.biominingbu.org/immunemir/, accessed on 18 September 2020)

Wang S. and coworkers presented a database, the ncRI, which collected experimentally validated ncRNAs in inflammatory disease from more than 2000 published papers. The current version of ncRI documents 11,166 manually curated entries that include 1976 miRNAs, 1377 lncRNAs, and 107 other ncRNAs, such as circRNAs (circular RNA), across 3 species (humans, mouses, and rats) [39]. Each entry in this database encompasses comprehensive information about ncRNA details and reference information. ncRI is an elaborate database and provides a comprehensive repository of ncRNAs and their roles in inflammatory disease that could be helpful for research on immunotherapy (freely available, accessed on 27 March 2021).

Jiang Y. and colleagues used an original approach to identify immune-related lnc RNAs, named IRlncRs, in renal cell carcinoma (RCC) from transcriptome RNA-sequencing data of RCC samples downloaded from the TCGA data portal. The Molecular Signatures database v4.0 specifies immune-related genes participating in immune processes, establishing the immune score using gene set enrichment analysis. Then, a Pearson correlation analysis was applied to correlate the immune score and the expression of lncRNA in the sequencing data of RCC patients. Moreover, clinical data about these patients were also downloaded to extract the overall survival (OS) and excluded patients with OS ≤ 30 days. IRlncRNAs associated with clinical outcome were selected by univariate Cox analysis using R software survival packages (*p* < 0.01), and hazard ratio was used to include survival-related IRlncRNA (sIRlncRNAs) into both protective and deleterious portions. These were used to develop the immune-related risk score (IRRS). Using the described workflow, the authors identified 7 sIRlncRNAs associated with RCC prognosis, and the top three (ATP1A1-AS1, IL10RB-DT, and MELTF-AS1) were included to build the IRRS model and to divide the RCCs into high- and low-risk groups. Finally, the authors concluded that the risk-evaluating scores based on the three sIRlncRNA signatures can contribute to identifying high-risk patients from patients with the same clinical and molecular characteristics and can allow for individualized and appropriate therapeutic strategies [40].

A similar approach was used by Xia P. and colleagues using TCGA and Chinese Glioma Genome Atlas (CGGA) patients. In this study, 812 immune-related lncRNAs were specifically associated with glioma. By Cox regression and LASSO analysis, the authors constructed a risk score formula to explore the different OSs between the high- and low-risk groups. Eleven immune-related lncRNAs were correlated with survival and included in the risk score (RS) formula. The authors observed that glioma patients with a high-risk score held poor survival in both the TCGA and CGGA groups. In fact, the RS formula could effectively predict the prognosis of glioma patients (5-year AUC- Area Under the Curve- = 0.749) and showed high prediction accuracy in the CGGA dataset (5-year AUC = 0.730). This is the success of the model building with powerful predictive function, which provides certain guidance values to the analysis of glioma pathogenesis and clinical treatment and allows users to identify potential therapeutic targets for glioma treatment [41].

Li Y. and colleagues introduced an integrated algorithm, ImmLnc, for identifying lncRNA regulators of immune-related pathways. They comprehensively charted the landscape of lncRNA regulation in the immunome across 33 cancer types and showed that cancers with similar tissue origin are likely to share lncRNA immune regulators. Moreover, immune-related lncRNAs are likely to show expression perturbation in cancer and are significantly correlated with immune cell infiltration. The authors applied ImmLnc to identify three molecular subtypes (proliferative, intermediate, and immunological) of non-small cell lung cancer (NSCLC). These subtypes are characterized by differences in mutation burden, immune cell infiltration, expression of immunomodulatory genes, response to chemotherapy, and prognosis. The ImmLnc pipeline used by authors, supported by the resulting data, represents a valid tool to prioritize cancer-related lncRNAs and to serve as a valuable resource for understanding lncRNA function and to advance the identification of immunotherapy targets [42].

Using the data of gastric adenocarcinoma from The Cancer Genome Atlas (TCGA), Chen T. and coauthors developed and validated a lncRNAs model for automatic microsatellite instability (MSI) classification using a machine learning technology: support vector machine (SVM). The C-index was adopted to evaluate its accuracy. Using the SVM, a lncRNAs model was established, consisting of 16 lncRNA features. In the training cohort with 94 gastric cancer (GC) patients, accuracy was confirmed with AUC 0.976 (95% CI, 0.952 to 0.999). Accuracy was also confirmed in the validation cohort (40 GC patients) with AUC 0.950 (0.889 to 0.999). Moreover, a high predicted score was correlated with better disease-free survival (DFS). The prognostic values of overall survival (OS) and DFS were also assessed in this model in the patients with stages I–III and lower OS with stages I–IV. In conclusion, the authors demonstrated that the identification of 16 lncRNA signatures was able to classify MSI status. The correlation between lncRNAs and MSI status indicates the potential roles of lncRNAsin immunotherapy for GC patients. Nevertheless, the pathway of these lncRNAs, which might be a target in anti PD-1/PD-L1 immunotherapy are needed to be further study [43].

Despite some progress and the development of different tools, listed in Table 2, able to correlate lncRNAs and immune-response, more work remains to be done to gather the immune and inflammatory disease-related factors that play important roles in the immune system and may represent novel potential targets for immunotherapy.

## 4. ncRNA in Immune Escape

Despite the promising role of immunotherapy in the fight against cancer, tumor immune-escape (TIE) remains the most critical challenge to overcome. TIE is the last phase of the cancer immunoediting process during which tumor cells acquire the ability to avoid immune system recognition through several strategies such as (i) reduced immunogenicity, which consists in loss of tumor-associated antigens or alteration of the antigen presentation mechanisms; (ii) deregulation of cell metabolism and cytokine production; (iii) aberration in immunosuppressive cells; and (iv) upregulation of immune checkpoints [1]. Recently, the relationship between ncRNAs and the TIE mechanism in cancer has been extensively elucidated and schematically represented here in Figure 1 [44].

### 4.1. ncRNAs Regulate Tumor Antigen Presentation

One of best-known mechanism of TIE is the loss in cancer immunogenicity that can arise from the elimination of antigenic tumor clones and/or an alteration in antigen processing and presenting machinery [45]. Class I MHC is composed of polypeptide sequence associated with β2 microglobulin, expressed by all nucleated cells. Endogenous peptides presented in MHC-I complex are processed in immunoproteasome, loaded in peptide loading complexes (PLCs) and translocated into the endoplasmic reticulum (ER) associated with TAP1-TAP2 and other proteins such as tapasin and calreticulin. Peptides are further edited by endoplasmic reticulum aminopeptidase1 (ERAP1) and loaded onto the MHC-I complex. If the affinity of the peptide for MHC-I is high, this complex will be transported first into the Golgi apparatus and then on the cell surface [46]. Furthermore, exogenous peptides, commonly presented in the MHC-II complex to TCD4+ lymphocytes, can be cross-presented to TCD8+ lymphocytes loaded onto the MHC-I complex in antigen presenting cells (APC). Defects in antigen processing, presentation, and recognition are common during cancer progression, and both miRNAs and lncRNAs are involved in regulation of this pathway. For example, miR-27a acts as an oncomiR through downregulation of MHC-I expression affecting tumor progression. Consistently, tumors with high miR-27a and low MHC-I levels are associated with poor prognosis [47]. Mari L. et al. identified miR-125a as a regulator of MHC-I expression in esophageal adenocarcinoma cells via direct binding of 3′ UTR TAP2 mRNA and found that miR-125a level inversely correlates with TAP2 and MHC-I expression both in adenocarcinoma and nontumor cells [48]. Lazaridou M. and collaborators identified miR-26-5p and miR-21-3p involvement in the immune escape mechanism mediated by MHC-I. They discovered that overexpression of miR-26b-5p and miR-21-3p induces the downregulation of TAP1 with consequential reduction in the expression of HLA class I cell surface antigens, leading to impairment in T-cell recognition [49].

### 4.2. Role of ncRNAs in Tumor Metabolism

Reprogrammed energy metabolism has emerged as a new hallmark of cancer, and miRNA and lncRNA play key roles in its regulation. Unlike normal cells producing energy using mitochondrial oxidative phosphorylation, cancer cells satisfy energy needs to sustain uncontrolled proliferation via glycolysis followed by lactate acid fermentation in a process known as the Warburg effect [50]. Lactate production, through TME acidification, leads to immune cell dysfunction and alteration in cytokine production. NcRNAs regulate expression and function of glucose transport, enzymes, and transcription factors involved in aerobic glycolysis. Wang Y. et al. demonstrated that lncRNA-p23154 binds miR-378a-3p, which represses Glut1 expression by targeting its 3′UTR by promoting invasion and metastasis in oral squamous cell carcinoma [51]. In ovarian cancer, it has been demonstrated that the lncRNA LINC00504 stimulates aerobic glycolysis via the downregulation of miR-1244, which is involved in the regulation of glycolysis-related enzymes [52]. In addition to glycolytic metabolism, amino acid pathways are also deregulated in cancer cells. The nonessential amino acid (AA) glutamine represents the main nutrient source in sustaining uncontrolled cancer proliferation and many tumors are highly dependent on glutamine metabolism [53]. Several miRNAs and lncRNAs regulate the glutamino-lysis pathway in cancer at different levels. For example, miR133a-3p blocks the glutamine metabolism in gastric cancer, targeting gamma-aminobutyric acid receptor-associated protein-like 1 (GABARAPL1) and inhibiting autophagy, a process by which gastric cancer cells recycle glutamine [54]. Indoleamine 2,3-dioxygenase (IDO) and tryptophan 2,3-dioxygenase (TDO) are expressed in tumor cells and promote immunosuppression via the recruitment of immunosuppressive cell subsets, such as T regulatory cells (Tregs) and myeloid derived suppressor cells (MDSC) and trough secretion of inhibitory cytokines and growth factors (IL-6, IL-10, and TGF-β) [44]. Metastasis-associated lung adenocarcinoma transcript-1 (MALAT-1) is one of the first oncogenic lncRNA discovered and associated with metastasis in early-stage NSCLC. This lncRNA plays its tumorigenic activity via alternative splicing, transcriptional regulation, epigenetic modification, and miRNA sponge and has been recently investigated for its immunosuppressive role, exerted by promoting M2 macrophage polarization and by inducing IDO in mesenchymal cells [55].

### 4.3. ncRNAs as Crucial Players in TME

For many years, the primary goal of cancer research was the identification of genetic mutations in cancer cells only, disregarding surrounding microenvironment. However, the relevance of the TME counterpart is now emerging. Tumor cell growth is supported by a complex-surrounding niche that includes fibroblasts, endothelial cells, and innate and adoptive immune cells nestled in the extracellular matrix (ECM). All these cell populations secret cytokines, chemokines, growth factors, and metabolites within the niche, which in turn may modify the oxygen level and the pH. This complex network of interactions supports tumor onset and progression. Stromal cells and immune cells are the leading players in the TME scenario in which signaling molecules induce antitumor or protumor effects depending on the reciprocal regulation and the dynamic crosstalk among all the cell types involved in the niche. Although there are many differences within the TME composition between patients, tumors can be characterized by immune cell density and inflammation such as hot tumors, immunosuppressed tumors, excluded immune tumors, or cold tumors. Hot tumors are characterized by a high density of cytotoxic T cells, by a high level of activation/exhaustion markers (PD1, TIM3, and LAG3), and frequently by genomic instability. Immunosuppressed tumors are instead characterized by high infiltration of immune-suppressive cells such as MDSCs and Tregs secreting immune-suppressive factors (TGFβ, IL-10, and VEGF- vascular endothelial growth factor-). The excluded immune class includes tumors in which there is an imbalance between stromal and immune components of TME, without T cell tumor infiltration, and with altered genetic and epigenetic TME regulatory pathways. Finally, cold tumors are characterized by the absence of T cells in the tumor bed and by defects in the antigen presentation machinery and T cell-mediated killing [56]. Dissecting the TME composition and understanding its complex regulatory network, it may significantly impact therapeutic planning and patient prognoses.

In general, the role of ncRNAs in the immunosuppressive TME has been underlined by several reports [57,58,59,60]. Among the cellular population of the microenvironment, we mention cancer-associated fibroblasts (CAFs), tumor-infiltrating lymphocytes (TILs), tumor-associated macrophages (TAMs), Tregs, and Th17 lymphocytes by highlighting recent literature concerning the role of ncRNAs in its regulation [61]. CAFs include the stromal cells that are activated in response to TGF-β released from tumor cells; are involved in inflammation, epithelial to mesenchymal transition (EMT), and angiogenesis; and secrete several growth factors (such as HGF-hepatocyte growth factor-, IGF-insulin-like growth factor, VEGF, EGF-epidermal growth factor, and PDGF-platelet-derived growth factor), cytokines (such as IL-1, IL-6, and IL-8), and enzymes (such as matrix metalloproteinases) [62]. Li P. and colleagues demonstrated the miRNA-mediated regulation of crosstalk between cancer cells and CAFs by highlighting the role of miR-149 in PGE2 and IL-6 signaling [63]. On the other hand, Zhang Y. et al. focused on the downregulation of miR-101 in CAFs with consequent upregulation of CXCL12, enabling lung cancer cells to proliferate, migrate, and invade [64]. Chatterjee A. and collaborators identified the upregulation of miR-222 in CAFs with respect to normal fibroblasts (NFs) and observed that the inhibition of miR-222 could impair the CAF-dependent progression of breast cancer cells [65]. Santolla M.F. et al. demonstrated that LNA-i-miR-221, a novel antisense oligonucleotide (ASO), was developed to specifically inhibit miR-221 oncogenic activity [66,67,68,69,70,71] and reverts the A20 downregulation and upregulation of c-Rel induced by miR-221 in breast cancer cell models. Moreover, the authors established that the miR-221-dependent recruitment of c-Rel to the NF-kB binding site located within the CTGF promoter region is prevented by using LNA-i-miR-221 that specifically downregulates CTGF mRNA and protein levels and silencing c-Rel. Finally, that cell growth and migration induced by miR-221 in MDAMB 231 and SkBr3 breast cancer cells as well as in CAFs are abolished by LNA-i-miR-221 and silencing c-Rel or CTGF [72]. Furthermore, in animal models, LNA-i-miR-221 exerts an anti-inflammatory effect reducing IL-8, MCP-1, and IL-6 plasma levels during treatments whereas no changes in TNF-α, IFN-γ, and IL-4 could be detected. Contrariwise, the downregulation of miR-214 in CAFs compared to NFs has been described as indicative of EMT FGF9-mediated in gastric cancer cells [73]. Colvin E. K. et al. described a signature of lncRNAs in ovarian CAFs, which are linked to poorer survival [74], while Liang Ding et al. identified a stromal lncRNAs signature that promotes oral squamous cell carcinoma progression by reprogramming NFs to CAFs in an IL-33-dependent manner [75]. Particular attention has been paid to tumor-infiltrating lymphocytes (TILs) as well as tumor-associated macrophages (TAMs) because of their strategic role influencing the balance between pro- and antitumorigenic effects. Zarogoulidis P. et al. highlighted the role of ncRNAs in the TME by revealing the function of miR-155 as an immune system activator, able to promote TIL infiltration [76].

Tregs are a regulatory subset of the CD4+T lymphocyte characterized by the expression of Foxp-3. High Tregs infiltration is associated with poor prognosis in many types of cancer [77]. Tregs accumulate in TME and are attracted by chemokines (CCL17, CCL22, and CCL1) and exert negative regulation on T lymphocytes and on dendritic cells through the secretion of immunosuppressive cytokines and growth factors such as IL-10 and TGF-β and the expression of co-inhibitory receptors (PD-1 and CTLA4) [78]. Among all investigated miRNAs, miR-155, miR-181, and miR-17-92 play key roles in Treg differentiation and function [79]. Recently, the lncRNA Flatr has been identified as a key regulator in Tregs switch, promoting Foxp-3 expression in vitro [80]. In multiple myeloma (MM) microenvironments, the direct interplay between bone marrow stroma cells (BMSCs) and plasma cells (PCs) plays a crucial role in disease progression and skeletal destruction. Specifically, BMSCs interact with MM cells and support MM proliferation and osteoclast (OCL) activity and inhibits bone formation. Pitari M. et al. demonstrated that miR-21 target the 3′UTR of OPG (osteoprotegerin), inducing severe imbalance in the RANKL (Receptor Activator of Nuclear Factor κ B ligand)/OPGratio, which is the main driver of bone homeostasis and that OCL activity [81]. Th17 is a specific subset of CD4 T cells characterized by higher secretion of IL-17 and other inflammatory cytokines and plays a dual role in tumor promotion and suppression. In the MM microenvironment, Th17 sustains MM cells growth and osteoclast-dependent bone damage [82]. In this context, miR-21 skews Th17 differentiation to the Th1 phenotype, leading to a delay in MM growth and an attenuation of bone disease [83].

TAMs are macrophage subpopulations deeply involved in tumor-promoting inflammation in which ontogenesis is still under debate. In fact, although initially TAMs were considered to exclusively originate from monocyte precursors recruited to the tumor site by chemokines/chemokine receptors axis, recent studies highlight the role of ResMac (resident macrophages) arising from yolk sac or fetal liver-derived progenitors and characterized by stem cell-like behavior [84].

TAMs play a central role in TME, since they promote tumor growth, angiogenesis, metastasis, tissue shaping, immune-reprogramming, and drug-resistance. Moreover, high TAM infiltration is associated with poor prognosis in several malignancies.

A growing body of information is available on the role of ncRNA in TMA polarization and regulation [85]. Zhou L. et al. proposed lincRNA-p21 as a crucial regulator of TAMs function in promoting breast cancer progression. In fact, lincRNA-p21 knockdown induces the polarization of macrophages in the M1 pro-inflammatory phenotype via the MDM2/p53 and NF-κB/STAT3 pathways [86]. Chen C. et al. demonstrated that LNMAT1 is upregulated in node-positive bladder cancer and is involved in bladder lymphatic metastasis. Specifically, LNMAT-1 promotes CCL2 upregulation and induces macrophage recruitment into bladder tumors, which leads to VEGF-C upregulation, resulting in lymphatic metastasis [87].

Frank A. et al. reported a high miR-375 level in breast cancer cells, which promote macrophage recruitment via the CCL2 axis. Moreover, the authors found that miR-375 is released during apoptosis by breast cancer cells and transferred to TAMs. In macrophages, miR-375 uptake is facilitated by a CD36 receptor and induces TAM infiltration in TMEs through direct targeting of key regulators of cell migration [88].

### 4.4. ncRNA and Immune Checkpoint

Physiologically, T cell activation requires three different signals: (i) interaction between TCR and MHC, (ii) binding of co-stimulatory molecules expressed on APC surfaces and relative receptors expressed on T-lymphocytes, and (iii) secretion of cytokines that induces T cell proliferation and expansion [89]. The immune system maintains a balance between T cell activation and inhibition in order to achieve homeostasis with nonredundant synergic stimulatory and inhibitory pathways known as immune checkpoint [44].

Recently, immune checkpoint deregulation has emerged as a crucial mechanism of cancer immune resistance and immune checkpoint therapy represents a powerful approachto enhance antitumor response. After the discovery of CTLA-4 and PD-1, several other targets for immune checkpoints such as TIM-3, LAG3, TIGIT, VISTA, BTLA, Siglec-15, and B7-H3 have been identified, and novel monoclonal antibodies or other blocking molecules have been developed [90].

ncRNAs are involved in a variety of cell pathways and in immune checkpoint regulation. CTLA-4 is a trans-membrane molecule member of the immunoglobulin-related receptor family expressed on CD4+ and CD8+ T lymphocytes able to inhibit T cell signaling by binding with CD80 and CD86, preventing CD28 engagement and, subsequently, PI3K activation and Zap70 formation. To date, two miRNAs targeting CTLA-4 have been identified. MiR-138 acts as a tumor suppressor in many types of cancer and, through its binding at the 3′UTR of CTLA-4 and PD-1, reduces glioma cell growth in vivo by suppressing immune checkpoint expression in human CD4+ T lymphocytes [91]. According to Huffaker TB et al., miR-155 promotes anticancer immune responses via the inhibition of CTLA-4 expression on T lymphocytes, and its overexpression may improve immunotherapy [92].

PD1 is an inhibitory receptor expressed on T-activated lymphocytes and on B cells and works as an adoptive immune break in order to maintain peripheral tolerance and to prevent T cell exhaustion [93]. Specifically, PD1 through its binding with PDL-1 and PDL-2 inhibits TCR downstream signaling by recruiting SHP2 or SHP1 phosphates, inhibits pro-inflammatory cytokines release, and promotes T cell apoptosis via downregulation of antiapoptotic molecules [94].

Several ncRNAs involved in PD1/PDL-1 axis regulation have been recently identified. In NSCLC models, for example, p53 modulates PDL-1 via miR-34, which directly binds to the *PDL1* 3′UTR [95]. Mastroianni J. et al. found that miR-146a is upregulated in the melanoma TME and regulates IFNY-STAT1, leading to increased PDL-1 levels. Starting from this premise, they proposed a novel immune therapeutic approach based on anti-PD1 and miR-146a antagomir. They demonstrated that this combined treatment synergically enhances the anticancer immune response in melanoma mouse models [96]. The lncRNA AFAP1-AS1 is co-expressed with PD-1 lymphocytes associated with nasopharyngeal carcinoma, resulting in aggressive and poor prognosis phenotype. Although the molecular mechanism has not yet been fully clarified, AFAP1-AS1-directed epigenetic modification by PDC1 binding has been hypothesized [97].

T cell immunoglobulin and mucin-domain containing-3 (TIM-3) is a co-inhibitory receptor expressed by several immune cells such as IFN-y-secreting lymphocytes, myeloid cells, NK, macrophage, and dendritic cells. Upon Galeactin 9 or CEACAM1 binding, TIM3 exerts an inhibitory function, inducing T cell anergy and TCR suppression, inhibiting Th1 response and IFNγ and TNFα secretion, suppressing innate immune-response, and disrupting the immunological synapse [98]. To date, several ncRNAs involved both in inhibition and activation of TIM-3 have been identified. For example, in hepatocellular carcinoma (HCC), which is a hot tumor, the lncRNA Tim3 mediates T cell exhaustion and inhibits T cell immune response, leading to tumor immune suppression by TIM-3-specific binding [99]. B and T lymphocyte attenuator (BTLA) is an inhibitory receptor expressed on activated Th1cells but not on Th2 lymphocytes. Crosslinking BTLA with antigen receptors induces its phosphorylation and association with SHP-1 and SHP-2 phosphatase and leads to a lower IL-2 production [100]. According to Liu J. et al., miR-155 is upregulated in T-activated lymphocytes and specific targets the 3′UTR of BTLA, decreasing BTLA surface expression by about 60% [101].

Due to a recent discovery of other immune checkpoints, such as LAG3, VISTA, Siglec-15, and B7-H3, ncRNAs have been yet identified as a regulator in these inhibitory pathways, and efforts will have to be made in order to shed light on ncRNA–immune checkpoint interactions.


## 5. ncRNAs in Immunotherapy Resistance

Drug resistance represents, until now, an obstacle in the efficacy of treatment with major effects in disease relapse/progression and prognosis. Several data from clinical studies in patients treated with T-cell-based immunotherapy demonstrated that at least 30–50% of cancers after an initial response develop a primary or secondary resistance [93,102]. The potential cause of the onset of resistance to immunotherapy is due to immune evasion from immune-surveillance, through tumor cells and TME alterations, at different levels [103]. Some lncRNAs, named immune-related lncRNAs [104], play an important role via differential regulation of the T-cell-mediated immune response and inflammatory cytokines release, resulting in tumor immunosuppressive TME, and take advantage of immune checkpoint pathways. Recent evidence suggests the potential therapeutic role of lncRNAs modulation as an immune sensitizer to overcome immunotherapy resistance. For example, preclinical data showed that the inhibition of nuclear-enriched autosomal transcript1 (NEAT1), a lncRNA associated with immunosuppression, can attenuate CD8+ T-cell apoptosis with an increase in cytolytic activity mediated by the miR-155/Tim-3 pathway and subsequent enhancement of the immune activity [105]. Among lncRNAs affecting antigen presentation, a higher expression of long intergenic noncoding RNA for kinase activation (LINK-A), found in a percentage (25%) of triple-negative breast cancer patients, seems to regulate negatively the recruitment of APC and CD8+ T cells with a low infiltration of APCs and activated CD8+ T cells as well as the β-2M and MHC-I expression, which was found decreased also [106,107]. The prognostic role of LINK-A may be due to its action on the degradation of TPSN, TAP1, TAP2, and CALR proteins of the peptide-loading complex (PLC) with an effect on the loading and editing of MHC-I. Therefore, LINK-A inhibitors can potentiate the effect of ICIs with an increment at the tumor level in the infiltration of hyperactivated CD8+ T cells [106]. Moreover, lncRNAs such as HOX transcript antisense intergenic RNA (HOTAIR) and MELOE have roles as suppressors of antigen presentation, thereby promoting immune evasion [108,109]. In diffuse large B-cell lymphoma (DLBCL), MALAT1 upregulates, through miR-195, the expression of PD-L1 and promotes migration and immune escape, with CD8+ T cells mediated. The inhibition of MALAT1 could revert this effect [73], and inhibition of the MALAT1 interaction with miR-101 modulates cisplatin and temozolomide resistance in lung cancer and glioblastoma, respectively [110,111]. Several reports confirm the role of other lncRNAs in the recruitment and activity of immunosuppressive cells, such as MDSCs and Tregs, for which presence in the TME confers a worse prognosis and immune therapy resistance [112]. MDSC differentiation into monocytic (Mo-) MDSCs is mediated by the pseudogene lncRNA Olfr29-ps1 highly expressed in MDSCs and with suppressive activities. The interaction between lncRNA Olfr29-ps1 and the miR-214-3p modulates the transformation of MDSCs through the N6-methyladenosine (m6A) modification via IL6, which also enhances Olfr29-ps1 expression [113]. Moreover, lnc-chop regulates the function and differentiation of MDSCs in tumors where the inhibition of lnc-chop in MDSCs increases the release of IFN-γ by CD4+ and CD8+ T cells through induction of the immune suppressive environment. Activation of the transcription factor CCAAT-enhancer-binding protein β (C/EBPβ) and upregulation of the expression of Arg-1, NOS_2_, NOX_2_, and COX_2_ via binding to both the C/EBPβ homologous protein (CHOP) and the liver-enriched inhibitory protein (LIP) play important roles in the modulation of this process. Additionally, lnc-chop increases NO, H_2_O_2_, and ROS production and the expression of Arg-1 by promoting the enrichment of the histone H3 lysine 4 trimethylation (H3K4me3) in the promoter region of Arg-1, NOS_2_, NOX_2_, and COX_2_ [114]. Instead, the effect of lncRNAs on the differentiation and distribution of Tregs is dual. Lnc-Smad3 and H3K4 methyltransferase Ash1l exert contrasting effects in the polarization of Tregs. In fact, the Foxp3 locus is regulated differently. Smad proteins are activated by TGF-β through phosphorylation and then the Smad complex binds to the Foxp3 locus, inducing its expression, which polarizes Treg cells [115]. Xiong G. et al. demonstrated that linc-POU3F by binding of and TGF-β activation of the TGF-β signaling pathway promote Tregs differentiation and distribution in gastric cancer [116].

## 6. Role of Exosomal ncRNA as Anticancer and Drug Resistance Cargo

Exosomes are lipid bilayer vesicles, with a size range of 40–150 nm in diameter, that are released via exocytosis into the extracellular space by a variety of cell types including immune cells and cancer cells. These nanovesicles can shuttle a plethora of molecules characterized by parent cells including carbohydrates, lipids, proteins, and nucleic acids [117]. It is known that exosomes interact with other cells, transferring their cargo, suggesting that they could modify the phenotype and genetic profile of recipient cells. Several reports have also reported that exosome level is correlated with tumor stage and metastasis [118]. In addition, tumor-derived exosomes (TEX) can promote cancer progression via modification or suppression of the immune response and can induce therapy resistance. These actions can be carried out by TEX antigen-presenting properties or cargo transferring. For example, TEX derived from oral squamous cell carcinoma and histiocytic lymphoma could induce immunosuppression through the CD95 (Fas) receptor and FasL+ exosomes signaling on activated CD8+ T cells [119]. TEX derived from LLC(Lewis lung cancer) and 4T1 breast cancer cells inhibit the differentiation of myeloid precursors into dendritic cells and induce DCs (dendritic cells) apoptosis [120]. One of the mechanisms by which tumor cells evade immune surveillance is via upregulation of PD-L1 expression on the cell surface. Growing experimental evidence indicates that exosomes are enriched with PD-L1 and that this extra-tumoral receptor has been implicated as a mechanism of resistance in immunotherapy, as represented in Figure 2. Specifically, immunosuppression is realized when exosome PDL-1 binds to anti-PD-L1, leaving the tumor PD-L1 exposed, or when exosome PDL-1 binds to PD-1 on effector T cells despite monoclonal antibody treatment [121].

In a similar way, B-cell lymphoma exosomes are enriched in CD20, inducing resistance to rituximab, the first therapeutic monoclonal antibody approved for CD-20+ B cell malignancies [122]. Another mechanism to generate the exosome-mediated drug-resistant phenotype is the transmission of nucleic acids. TEX can transport various ncRNAs that are able to promote immunosuppression. For instance, Liu S. et al. demonstrated that, in the endoplasmic reticulum of stressed HCC cells, the transfer of tumor exosomal-miR-23a-3p to macrophage induces PI3K-AKT pathway activation by PTEN inhibition, increases PD-L1 expression, and inhibits of T-cell function [123]. Similarly, in tumor-derived serum exosomes from nasopharyngeal carcinoma patients or TW03 cell lines, the level of five exosome miRNAs (hsa-miR-24-3p, hsa-miR-891a, hsa-miR-106a-5p, hsa-miR-20a-5p, and hsa-miR-1908) were significantly higher than in normal counterparts [124].

TEX-derived lncRNAs were also found to be associated with immunosuppression and TIE. The expression level of exosome ZFAS1 and MALAT-1 were highly expressed in gastric cancer and NSCLC patients, respectively, and both showed correlations with tumor stage and lymphatic metastasis [124,125,126]. In bladder cancer, hypoxia induces upregulation of lncRNA UCA1 in both cancer cell line and TEX, promoting cancer cell migration and invasion [127].

Ni C. et al. demonstrated that breast cancer-derived exosomes transmit lncRNA SNHG16 to Vδ1 T cells, which act as competing endogenous RNAs via sponging miR-16-5p and leads to de-repression of miR-16-5p targets, with SMAD5 among these. Activation of the SMAD5 pathway stimulates CD73 expression, which promote immunosuppression via adenosine generation [128]. Moreover, Liang Z. et al. have shown that lncRNA RPPH1 promotes colorectal cancer metastasis by interacting with TUBB3 and by promoting exosomes-mediated macrophage M2 polarization [129]. Another example is the lncRNA LINK-A that was found upregulated in triple negative breast cancer patients resistant to PD-1 blockade drugs [106]. Immune cells also produce and release exosomes, which are able to influence the TME. For example, Treg cell exosomes miRNA (let-7d) strongly inhibited Th1 cell activity by inhibiting COX-2-mediated IFN-γ production [119]. In colorectal cancer cells, M2 macrophage-related exosomes stimulate tumor cell migration and invasion transferring miR-21-5p and miR-155-5p, which target and downregulate BRG1, a key factor promoting colorectal cancer metastasis [130].

**Table 1 cancers-13-01587-t001:** miRNAs and lncRNAs involved in cancer immunoediting, tumor cell and microenvironment (TME) modulation, and immunotherapy resistance.

Regulatory Function	ncRNA Name		Target Name	Target Modulation	Tumor Type	Ref.
	miRNAs	lncRNAs				
**Tumor antigen presentation**	**miR-27a**		**TAP2**	Downregulation of MHC-I expression	Esophageal adenocarcinoma	[47]
	miR-26-5p and miR-21-3p		TAP1	Downregulation of TAP1 and reduced expression of HLA class I cell surface antigens	Melanoma	[49]
**Tumor metabolism**		lncRNA p23154	miR-378a-3p	Repression of Glut1 expression through miR-378a-3p binding to the UTR of the gene	Oral squamous cell carcinoma	[51]
		lncRNA LINC00504	miR-1244	Stimulation of aerobic glycolysis through PKM2, HK2, and PDK1	Ovarian cancer	[52]
	miR133a-3p		GABARAPL1	Blockade of glutaminolysis by the reduction of the expression level of core enzymes including GLS and GDH	Gastric cancer	[54]
		MALAT-1	Promote VEGF expression not only through a direct pathway, but also through miRNAs, which deserves other studies	Immunosuppressive Properties of Mesenchymal Stem Cells (MSC) by Inducing VEGF and IDO	Mesenchymal stem cells	[55]
**Tumor** **Microenvironment**	miR-149		IL-6	Inhibition of the activation of tumor-promoting fibroblasts by the reduction of IL-6 expression	Gastric Cancer	[63]
	miR-101		CXCL12	Inhibition of the interaction between fibroblasts and cancer cells by the downregulation of CXCL12	Lung cancer	[64]
	miR-222		LBR	Downregulation of LBR by inducing normal fibroblasts to show the cancer-associated fibroblast (CAF) characteristics	Breast cancer	[65]
	miR-221		A20	Stimulatory action in breast cancer cells and in main component of the TME such asCAFs, through the involvement of A20/c-Rel/CTGF signaling	Breast cancer	[66,67,68,69,70,71]
		lincRNA-p21	P53	Direct targeting on p53, abolishment of MDM2 degradation to p53 by facilitating phenotype maintenance of Tumor-associated macrophages (TAMs)	Breast Cancer	[86]
		LNMAT1	CCL2	CCL2 upregulation, macrophage recruitment, and metastasis spreading	Bladder cancer	[87]
	miR-375	PXN and TNS3		Destabilization of PXN and TNS3, TAM infiltration	Breast cancer	[88]
	miR-21		OPG	OPG downmodulation and RANKL upregulation by playing a role in bone resorption/apposition balance	Multiple Myeloma	[81]
**Immune Checkpoint**	miR-138		CTLA-4 and PD-1	Inhibition of human checkpoint expression in Tregs. Downmodulation of CTLA-4, PD-1, and FoxP3 in CD4+ T cells	Glioma	[91]
	miR-155		IL7R	Repression of IL7R expression in response to activation signals by regulating T cell survival, homeostasis, and proliferation	Melanoma	[79]
	miR-34		PDL1	Downregulation of PDL-1	Non small cell lung cancer (NSCLC)	[95]
	miR-146a		IFNY-STAT1	Upregulation of PDL-1	Melanoma	[96]
		lncRNA AFAP1-AS1	PDC1	Upregulation of PD-1	Nasopharyngeal carcinoma	[97]
		lncRNA Tim3	TIM3	Binding of Tim3 and nuclear translocation of Bat3	Hepatocellular carcinoma	[99]
	miR-155		BTLA	Downregulation of BTLA surface expression	Tumor microenvironment	[101]
**Immunotherapy resistance**		NEAT1	miR-155/Tim-3	Downregulation of miR-155 and Tim-3 upregulation	Hepatocellular carcinoma	[105]
		MALAT1		Upregulation, through miR-195, of PD-L1 Inhibition of MALAT1 interaction with miR-101, modulation of cisplatin, and temozolomide resistance	Diffuse large B-cell lymphoma Lung cancer and glioblastoma	[72,110,111]
**ExosomalncRNAs** **As drug resistance cargo**		lncRNA Olfr29-ps1	miR-214-3p	Sponging of miR-214-3p and downregulation of miR-214-3p, which target MyD88 to modulate differentiation and function of MDSCs	Tumor microenvironment	[113]
		Lnc-chop	CHOP and the C/EBPβ isoform liver-enriched inhibitory protein	Activation of C/EBPβ, upregulation of arginase-1, NO synthase 2, NADPH oxidase 2, and cyclooxygenase-2	Tumor microenvironment	[114]
		lnc- POU3F3	TGF-β	Upregulation of TGF-β, distribution of Tregs in peripheral blood, enhance cell proliferation of gastric cancer	Gastric cancer	[116]
	miR-23a-3p		PTEN, AKT	PDL-1 upregulation in macrophages	Hepatocarcinoma	[123]
	hsa-miR-24-3p, hsa-miR-891a, hsa-miR-106a-5p, hsa-miR-20a-5p, and hsa-miR-1908		MARK1	Downregulation of the MARK1 signaling pathway	Nasopharyngeal carcinoma	[124]
		ZFAS1	D1, Bcl2, N-cadherin, Slug, Snail, Twist, Bax and E-cadherin	Upregulation of D1, Bcl2, N-cadherin, Slug, Snail, Twist, and ZEB1 and downregulation of Bax and E-cadherin	Gastric cancer	[125]
		MALAT-1	cyclinD1, cyclinD2 and CDK	Upregulation of cyclinD1, cyclinD2, and CDK; tumor growth promotion; migration; and apoptosis prevention in lung cancer cell lines	NSCLC	[126]
		lncRNA UCA1	E-cadherin, vimentin, MMP9 proteins	Decreasing of E-cadherin, increasing of vimentin and MMP9	Bladder cancer	[127]
		lncRNA SNHG16	acts as ce-RNA via sponging miR-16-5p	De-repression of miR-16-5p targets, SMAD5 among these	Breast cancer	[128]
		lncRNA RPPH1	TUBB3	Interaction with TUBB3 to prevent its ubiquitination, macrophage M2 polarization, metastasis spreading, and proliferation of colon cancer cells	Colorectal cancer metastasis	[129]
		lncRNA LINK-A	PtdIns (3,4,5) P3, inhibitory GCPRs, E3 ubiquitin ligase TRIM71	Enhancement of K48–polyubiquitination-mediated degradation of the antigen peptide-loading complex (PLC), and Rb and p53	Triple negative breast cancer	[106,107]
	miR-21-5p and miR-155-5p		BRG1	Downregulation of BRG1 leading to colorectal cancer cells migration and invasion.	Colorectal cancer metastasis	[130]

**Table 2 cancers-13-01587-t002:** Tools supporting an interactive integration of ncRNA players and targeting immune-related signature.

Algorithm or Signature ID	Freely Available	Description	Reference
ImmunemiR	http://www.biominingbu.org/immunemir/, accessed on 17 September 2020	repository for immune-related disease and miRNA associations	Prabahar A. et al. [35]
ncRI	http://www.jianglab.cn/ncRI/, accessed on 27 March 2021	comprehensive repository of ncRNAs and their rolesin inflammatory disease	Wang S. et al. [36]
IRlncRs	https://rdcu.be/ceXm7, accessed on 27 March 2021	immune-related risk score (IRRS) in RCC	Jiang Y. et al. [37]
11 immune-related lncRNAs signature	https://rdcu.be/ceXFC, accessed on 17 September 2020	immune-related lncRNAs for glioma risk score formula	Xia P. et al. [38]
ImmLnc	https://rdcu.be/ceXEP, accessed on 17 September 2020	integrated algorithm for identifying lncRNA regulators of immune-related pathways	Li Y. et al. [39]
16 lncRNAs signatures	https://rdcu.be/ceXGg, accessed on 17 September 2020	lncRNAs model for automatic microsatellite instability (MSI) classification using a machine learning technology	Chen T. et al. [40]

## 7. ncRNA-Based Therapeutic Approaches

ncRNA-based therapeutic strategies can be discussed if we consider miRNAs or lncRNA. miRNA-based therapies can be developed on two opposite strategies, to regulate miRNA aberrant expression via the replacement of downregulated miRNAs or the inhibition of upregulated miRNAs [131,132,133,134]. The replacement approach of miRNA aims to restore miRNA levels that are downregulated and/or selectively deleted in the cell. This strategy needs previous knowledge of the biologic function of the miRNA under investigation, and its involvement in the tumor growth suppression and progression, which specifically targets crucial mRNA involved in proliferation and survival of cancer cells. Reestablishing miRNA levels can be achieved by infection of viral vectors to stably express a specific miRNA [135] or by transient transfection of mature miRNAs, known as miRNA mimics, consisting in double-stranded oligonucleotides of 22-mer length approximately, bearing the same sequence of endogen mature miRNA or its precursor. To this aim, several types of delivery systems were developed, including polymeric vectors; lipid-based carriers; and positively and negatively charged or neutral, and inorganic materials [136]. On the other hand, miRNAs that are upregulated in cancer cells supporting proliferation and survival can be specifically inhibited based on targeting approaches, according to Watson–Crick base-pairing rules. To this aim, ASOs are extensively used as miRNA inhibitors (antagomirs), specifically designed to anneal in a complementary fashion to the “sense” miRNA strand, inducing RNAse-H-mediated degradation. ASOs may bring some modifications in the chemical structure to enhance the stability and affinity of the miRNA target. The most diffuse chemical modification introduced in the ASO are the modification in the 2-OH residue of ribose by the O-methyl (2 -OMe) or O-methoxyethyl (2 -MOE) group [137] and an extra bridge connecting the 2′ oxygen and 4′ carbon that locks the ribose in the 3′-endo conformation, enhancing base stacking and increasing hybridization properties. The latter characterizes the Locked Nucleic Acid (LNA) oligonucleotides and are mainly synthesized with a phosphorothioate (PS) backbone, where a sulfur atom substitutes a non-bridging oxygen atom, conferring resistance to nuclease degradation. LNA-PS oligonucleotides are highly soluble in water and stable in biofluids, with optimal bio-distribution within tissues and long-lasting knockdown of the target in vivo [68,138,139]. In addition to ASOs, other strategies have been developed to inhibit oncogenic miRNA functions: (i) miRNA sponges, transcripts that contain multiple tandem binding sites to a miRNA of interest, which acts as an miRNA decoy preventing their binding to target mRNA [140]; (ii) miR-MASKs, modified ASOs complementary to miRNA binding sites on the mRNA target [141], which mask the miRNA binding site and selectively inhibit the interaction of the miRNA with the specific mRNA target in order to antagonize repression. On the side of lncRNAs therapeutic strategies, besides the ASO strategy, which takes advantage from a DNA/RNA structure through RNA targeting by base pairing rules, a new class of ASOs have been developed for ncRNA inhibition strategies, named gap-mers. They are oligonucleotides containing LNA/DNA mixmers able to bind specifically the RNA target, forming an highly stable heteroduplex structure able to recruit the RNase-H enzyme because it brings a central “gap” of DNA nucleotides, inducing target degradation [142,143]. Target-specific genetic modification technology, i.e.,CRISPR (Clustered Regularly Interspaced Short Palindromic Repeats)/Cas9 genome editing, is currently used for specific DNA modification in targeted genes. Recent studies have found that CRISPR/Cas9 can successfully silence transcription of the lncRNA-expressing loci [144,145]. Different studies have found that more than 16,000 lncRNA promoters in the human genome could be targeted by guide RNAs [145]. For example, the knockout of lncRNA-NEAT1has been reported and lncRNA-MALAT1dramatically inhibited the metastasis of cancer cells [132,146,147]. To date, clinical application of the CRISPR/Cas9 system targeting lncRNA to treat cancer appears to be still elusive.

Aptamers, also called “nucleic acid monoclonal antibodies”, are single-stranded, highly structured DNA or RNA oligonucleotides that can bind to a wide variety of molecular targets, including proteins, peptides, DNAs, RNAs, small molecules, and ions, with high affinity and specificity, recently used as therapeutics. They have emerged as effective therapeutics for a wide range of human diseases, including solid and hematological tumors. As monoclonal antibodies (mAb), aptamers are able to fold in particular three-dimensional shapes and to bind specific targets but, differently from mAb, show low immunogenicity; have small size, increased chemical stability, and high-fidelity batch; and is easily produced. Quirico L. et al. developed an aptamer-based therapeutic tool for the inhibition of pro-metastatic miR-214 and reduction of metastasis by simultaneously overexpressing its downstream molecule, anti-metastatic miR-148b [148]. RNA or DNA aptamers are mainly developed and reported in the literature (see Kulabhusan P. K., Pharmaceutics, 2020, for review) [149] for their ability to target protein or cell receptors in addition to mAb, but the development of clinically useful aptamers for therapy is growing slowly compared to antibodies. Currently, the design of aptamers as carriers for cell-targeted delivery of ncRNA or inhibitors can be considered a promising new tool for intracellular delivery of “active” oligonucleotides conjugated to aptamers targeting cancer cells [150].

The inhibition of a lncRNA can also be reached by RNA interference (RNAi) technology. This is a biological process of specific gene knockdown via neutralizing targeted RNA by exogenous double-stranded RNA, which includes short interfering RNAs (siRNAs) and short hairpin RNAs (shRNAs). Despite its specificity, siRNA’s efficiency is transient due to its instability, while stem-loop shRNA may provide a durable and long-lasting effect in vivo [151,152]. Many reports use shRNAs to target lncRNAs in treating cancer, including lncRNA-BCAR4, HOTAIR, and lncRNA-PNUTS. These strategies may be used also to transfect exogenously synthesized lncRNA plasmids into cancer cells to up-regulate corresponding lncRNAs. However, while several studies support the above described therapeutic strategies, for siRNA several strategies, solid experimental data are still needed to shed light on the feasibility of this method.

Furthermore, indirect strategies could be employed to modulate ncRNA expression. By the use of selective small molecules inhibitors, identified by efficient screening of chemical libraries, the possibility to modulate the machinery that contributes to processes of specific miRNA maturation and degradation have been recently evidenced [153]. Conventional small-molecule compounds with broad structural diversities and drug-like physicochemical and PK properties are preferred entities to bind and manipulate highly structured RNA targets [154,155,156,157]. Similar to protein targets, macromolecule RNAs such as lncRNA are folded into highly structured entities for their interactions with small molecules [158]. Through complementary base pairings and other forms of physicochemical interactions, RNAs are folded into secondary, tertiary, and quaternary structures [159,160,161]; thus, small-molecule compounds have the potential to directly interact with unique higher-order structures (3D) rather than primary sequences [6].

Several clinical trials have evaluated miRNA-based therapeutic strategies for the treatment of cancer and other diseases [162,163,164]. The first miRNA mimics used as therapeutic, named Miravirsen (SPC3649), reached phase II clinical trials for the treatment of hepatitis C virus (HCV) infection. Miravirsen is an LNA-ASOs that blocks miR-122 interaction with HCV RNA, leading to virus destruction [165]. Then, MRX34, a liposome-formulated mimic of miR-34a, was investigated in a phase I clinical trial in patients with advanced solid tumors [166], showing relevant activity in hepatocellular carcinoma, renal cell carcinoma, and melanoma after intravenous (IV) infusion. However, multiple immune-related severe adverse events were registered and MRX34 development to phase II clinical trial was halted. Different preclinical studies have been performed to support the anticancer activity of miR-34a overexpression in different tumors [6,66,167,168,169,170,171]. miR-16 mimics have been also recently evaluated in a phase I clinical trial for patients with malignant pleural mesothelioma and advanced NSCLC that have failed standard therapy [172]. Bacterial-derived (EDV) packaging was used to delivery miR-16 mimics by IV infusion [173] conjugated with an EGFR-targeting antibody (TargomiRs). At the end of treatment, 27% of patients had progressive disease, 68% had stable disease, and 5% had a partial response [174]. Looking at miRNA inhibitors therapeutic strategies, MRG-106 (Cobomarsen), a synthetic LNA antimiR of miR-155, is presently investigated in an ongoing phase II trial to treat mycosis fungoides (a type of cutaneous T-cell lymphoma) and, in phase 1 trials for HTLV (Human T-lymphotropic virus) associated adult T-cell lymphoma/leukemia to diffuse large B-cell lymphoma and chronic lymphocytic leukemia. In the phase 1 trial for mycosis fungoides, the drug was delivered intratumorally, by subcutaneous injection, or by intravenous injection for 6 doses in the first 26 days and then weekly. Overall, treatments knocked down miR-155 and altered miR-155 target genes in patient biopsies compared with pre-treatment biopsies and were effective at reducing lesion burden and at improving quality of life. There were also no serious adverse events attributed to Cobomarsen in the trial and no evidence of immunosuppression over the course of almost two years [175]. Finally, a phase I clinical study (EudraCT: 2017-002615-33) is ongoing to assess the safety profile of a miR-221 inhibitor [69,70,176], LNA-i-miR-221, which is a second-generation phosphorothioate ASO, and will take advantages of LNA technology and PS backbone chemistry in terms of increased affinity to the target and resistance to nucleases [68,177]. In vitro and in vivo studies demonstrated that LNA-i-miR-221 exerts strong antitumor activity, specific inhibition of miR-221, and consequent modulation of its canonical targets, including p27Kip1, PUMA, PTEN, and p57Kip2, regulators of the cell cycle and apoptosis, providing evidence of its efficacy against multiple myeloma (MM) [67,71,178] and other tumors [72,179].

## 8. ncRNAs and Immune Checkpoint Inhibitors: Current Clinical Evidence and Future Perspectives

Immune checkpoint inhibitors (ICIs) are monoclonal antibodies targeting immune check points often upregulated on cancer cells and on surrounding immune and stromal microenvironment cells. ICIs have dramatically improved patient’s prognosis and are now approved, alone or in combination with other immunotherapeutic-based agents, chemotherapy, or target therapy, in 50 different cancer types.

From 2011, when the monoclonal antibody anti-CTLA (Ipilimumab) was first approved by the FDA (Food and Drug Administration) in metastatic and unresectable melanoma, ICIs have revolutionized the cancer therapeutic scenario. To date, 7 ICIs have received FDA approval and thousands of clinical trials are underway [180]. One of the most impressive achievements of ICI therapy has been long-term remission in some cancer types. Unfortunately, not all cancer patients benefit from ICIs, mainly due to the loss in tumor immunogenicity and to immune microenvironment phenotype. For these reasons, the dissection of mechanisms of primary and acquired resistance, the development of therapeutic strategies to overcome resistance, as well as the discovery of critical predictive biomarkers of ICIs response represent big challenges for cancer immunotherapy. In this context, the activity of ncRNAs has been recently investigated and several ncRNAs have been found to be correlated to ICIs response.

From a miRNA profiling study, seven miRNAs have emerged as predictive biomarkers in lung cancer patients treated with nivolumab (anti PD-1 mAb), although the underling molecular mechanisms have not been elucidated [181]. Sudo et al. identified miRNA related to nivolumab response in esophageal carcinoma patients, analyzing the serum miRNA level before, during, and after ICI therapy [182].

In a recently published cohort study, lncRNA-based immune subtypes associated with overall survival (OS) and response to cancer immunotherapy in 348 patients with bladder cancer and 71 patients with melanoma were investigated. Among the patients, four distinct classes with statistically significant differences in OS were identified. The greatest OS benefit was obtained in the immune-active class characterized by the immune-functional lncRNA signature and high T cytotoxic infiltration. This study also provided a lncRNA score for multi-omic panels in precision immunotherapy [183]. In another study, the lncRNA RP11705C15 played a key role in NSCLC immune response and was related to prognosis of patients treated with anti PD-1 immunotherapy [184].

Further studies on larger patient cohorts are required to validate ncRNA as predictive and prognostic biomarkers in patients treated with ICIs. Moreover, due to the pleiotropic role of ncRNA and the ability to selectively inhibit different checkpoint receptors simultaneously, RNA-based therapeutics may represent an exciting approach in cancer immunotherapy. Xu et al. demonstrate that miR-424 is inversely correlated with both PDL-1 and CD80 levels in ovarian cancer. On the other hand, the restoration of miR-424 induces T cell activation and reverse chemoresistance, suggesting a combination therapy of miR-424 mimics with ICIs [185].

The regulatory role of miR-28 in T cell exhaustion has been recently identified. Specifically, miR-28 inhibition led to an increase in three checkpoint inhibitor receptors at the same time (PD-1, TIM-3, and LAG3) and induced impairment of IL-2 and TNF-a secretion in T cells [186].

The lncRNA MIR-155HG was closely associated with overall survival (OS) of different tumor types such as cholangiocarcinoma, lung adenocarcinoma, skin cutaneous melanoma, head and neck squamous cell carcinoma, glioblastoma multiforme, kidney renal clear cell carcinoma, and glioma. The expression ofMIR155HGwas significantly correlated with infiltrating levels of immune cells, molecules, and immune checkpoint such as PD-1, PD-L1, CTLA4, LAG3, and TIM-3 [187]. Nowadays, targeting immune checkpoints with RNA therapeutics represents a promising approach in the cancer immunotherapy field. However, many efforts to investigate the molecular mechanism underlying the ncRNA and immune checkpoint association as well as significant improvements in drug development have yet to be made.

## 9. Conclusions

Experimental evidence clearly underlines the potential role of ncRNA therapeutics as a promising approach in the prime-time area of cancer treatment via the novel tools of immune oncology. Modern approaches to profiling and integrative analysis [188,189,190] allow for a depiction of the whole scenario of cellular and molecular interactions, while novel and emerging tools are now available for selective intervention. Breakthrough findings are expected in the next future.

## Figures and Tables

**Figure 1 cancers-13-01587-f001:**
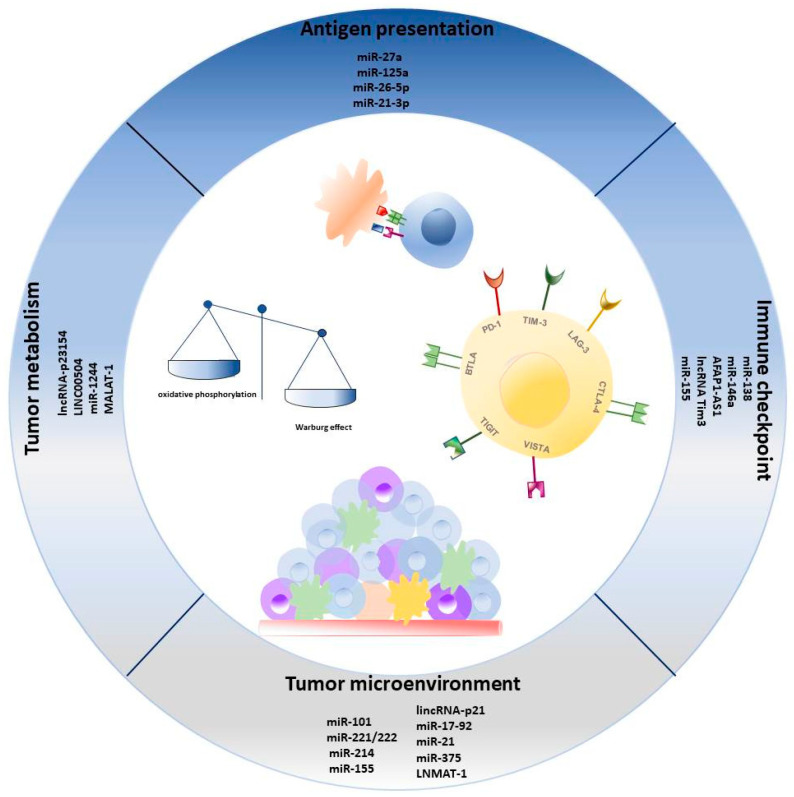
Overview of most relevant ncRNAs involved in cancer immune regulation of antigen presentation, tumor metabolism, immune checkpoint expression, and microenvironment composition.

**Figure 2 cancers-13-01587-f002:**
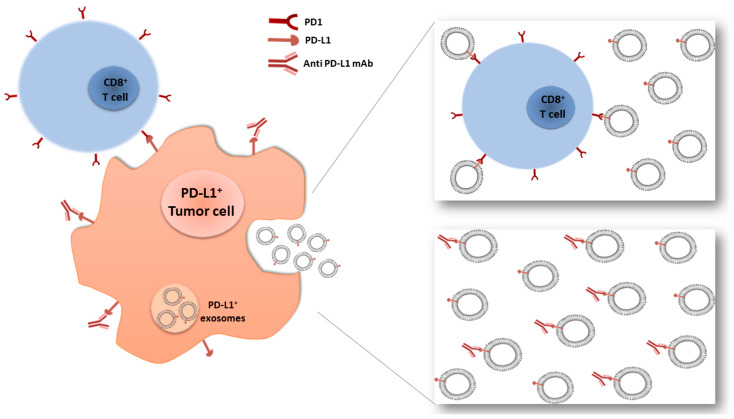
Exosomes-mediated immune-escape mechanisms. On the left is the mechanism of PD-L1+ tumor cell killing via PD1+/CD8+ T cell or anti-PD-L1 mAb binding. On the right are two possible mechanisms of acquired immunotherapy resistance mediated by PD-L1+ exosomes and secreted from the cancer cell: the direct binding between exosomes and T cells or between exosomes and anti PD-L1 monoclonal antibodies.

## Data Availability

Not applicable.

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
