# Peer review of "miRNAs and lncRNAs as Novel Therapeutic Targets to Improve Cancer Immunotherapy"

_cancers, 2021, doi:10.3390/cancers13071587_

Round 1
Reviewer 1 Report
This is a well-written review article focussing on the role of mirnas and lncrnas within the immune system in relation to the immune system. A minor concern related to the article is
- Since the article focusses primarily on mirnas and lncrnas, the title of the article should specifically mention that and not use an umbrella term of ncRNAs.
Author Response
Reviewer #1
Comments and Suggestions for Authors
This is a well-written review article focussing on the role of mirnas and lncrnas within the immune system in relation to the immune system. A minor concern related to the article is
Since the article focusses primarily on mirnas and lncrnas, the title of the article should specifically mention that and not use an umbrella term of ncRNAs.
We thanks the Reviewer for this suggestion and, accordingly, we changed the title: miRNAs and lncRNAs as novel therapeutic targets to improve cancer immunotherapy
Reviewer 2 Report
Comments for the authors
This review focused on ncRNAs as novel therapeutic targets to improve cancer immunotherapy.
Even if the immunotherapy had drastically improved the therapeutic approach on anti-tumor therapies, it is also true that non-responding patients lack new alternatives to overcome such a problem in many cases. The authors highlight the possible use of ncRNA as a supportive treatment combined with immunotherapy since ncRNA modulates many processes in cancer development and immune response evasion.
I found it is very fruitful that the authors summarize recent progress in such a very interesting field. However, the actual version of the manuscript raises some concerns, which should be addressed before publication.
General Comments
Comment 1:
It would be helpful to design a table to summarize the findings of the described ncRNA:
1) Name of ncRNA;
2) If it is described the lncRNA the target miRNA;
3) Target gene’s mRNA
4) the description of up or downregulated modulation of the mRNA;
5) effect on the function of the modulated gene
6) References.
Comment 2:
Please also add this important reference that could help in the interpretation of the naming of ncRNAs:
Seal RL, Chen LL, Griffiths-Jones S, Lowe TM, Mathews MB, O'Reilly D, Pierce AJ, Stadler PF, Ulitsky I, Wolin SL, Bruford EA. A guide to naming human non-coding RNA genes. EMBO J. 2020 Mar 16;39(6):e103777. doi: 10.15252/embj.2019103777. Epub 2020 Feb 24. PMID: 32090359; PMCID: PMC7073466.
Comment 3:
The effect of ncRNA on TAM is totally missing. in many solid tumors, up to 50% of infiltrating immune cells are macrophages. Moreover, the polarization effect of ncRNAs on macrophages is known and important also in tumor progression and prognosis. Please add one paragraph to cover this topic in “ncRNAs as crucial players in TME”.
Comment 4:
Line 88 – Please add in the sentence where the “Pre-miRNAs are generated”.
Comment 5:
Line 159 – Please define what does it mean “circRNA”.
Comment 6:
Line 423 – 424 - Please re-write the sentence to clarify better what the authors mean. It is not totally clear the description of the hypothetical mechanisms that modulate PD-1.
Comment 7:
Line 531 – Please explain that Rituximab is a monoclonal therapeutic anti-tumor antibody.
Comment 8:
Line 548 – Please change ceRNA with “competing endogenous RNAs (ceRNAs)”.
Additional comments
I would strongly suggest a deep check of written English because of many typos and sentences that seem not be as clear as they should be.
Line 35: “adoptive” instead of “adaptive”;
Line 485 “Lnc-chop” should be written with the lowercase letter “lnc-chop”.
Line 534: “endoplasmic reticulum stressed HCC cells” instead of “endoplasmic reticulum of stressed HCC cells”;
Line 602: “able to recruit RNase-H becouse brings …” instead of “able to recruit RNase-H enzyme because brings …”;
Line 638 – 641: Check spaces.
Font size problems
Line 431 “for example”;
Line 434 “(91)”;
Line 609 “lncRNA-NEAT1 and lncRNA-MALAT1”;
Line 631 “via”;
Line 635 “in vivo”;
Line 739 “of MIR155HG”
Author Response
Reviewer #2
Comments for the authors
This review focused on ncRNAs as novel therapeutic targets to improve cancer immunotherapy.
Even if the immunotherapy had drastically improved the therapeutic approach on anti-tumor therapies, it is also true that non-responding patients lack new alternatives to overcome such a problem in many cases. The authors highlight the possible use of ncRNA as a supportive treatment combined with immunotherapy since ncRNA modulates many processes in cancer development and immune response evasion.
I found it is very fruitful that the authors summarize recent progress in such a very interesting field. However, the actual version of the manuscript raises some concerns, which should be addressed before publication.
General Comments
Comment 1:
It would be helpful to design a table to summarize the findings of the described ncRNA:
1) Name of ncRNA;
2) If it is described the lncRNA the target miRNA;
3) Target gene’s mRNA
4) the description of up or downregulated modulation of the mRNA;
5) effect on the function of the modulated gene
6) References.
We thanks the Reviewer for this suggestion and, accordingly, we added a table numbered as tab.1
Comment 2:
Please also add this important reference that could help in the interpretation of the naming of ncRNAs:
Seal RL, Chen LL, Griffiths-Jones S, Lowe TM, Mathews MB, O'Reilly D, Pierce AJ, Stadler PF, Ulitsky I, Wolin SL, Bruford EA. A guide to naming human non-coding RNA genes. EMBO J. 2020 Mar 16;39(6):e103777. doi: 10.15252/embj.2019103777. Epub 2020 Feb 24. PMID: 32090359; PMCID: PMC7073466.
In the revised version of the manuscript, we added the reference as suggested.
Comment 3:
The effect of ncRNA on TAM is totally missing. in many solid tumors, up to 50% of infiltrating immune cells are macrophages. Moreover, the polarization effect of ncRNAs on macrophages is known and important also in tumor progression and prognosis. Please add one paragraph to cover this topic in “ncRNAs as crucial players in TME”.
We thanks the Reviewer for this important suggestion. In the revised manuscript, we included a section of ncRNA on TAM in the paragraph “ncRNAs as crucial players in TME”. [pag.9]
Comment 4:
Line 88 – Please add in the sentence where the “Pre-miRNAs are generated”. Done
Comment 5:
Line 159 – Please define what does it mean “circRNA”. Done
Comment 6:
Line 423 – 424 - Please re-write the sentence to clarify better what the authors mean. It is not totally clear the description of the hypothetical mechanisms that modulate PD-1.
As suggested by Reviewer, the sentence has totally re-wrote.
Comment 7:
Line 531 – Please explain that Rituximab is a monoclonal therapeutic anti-tumor antibody. Done
Comment 8:
Line 548 – Please change ceRNA with “competing endogenous RNAs (ceRNAs)”. Done
Additional comments
I would strongly suggest a deep check of written English because of many typos and sentences that seem not be as clear as they should be.
We thanks the Reviewer for this suggestion. We check the written English for typos and some sentences have been totally re-wrote.
Line 35: “adoptive” instead of “adaptive”; Done
Line 485 “Lnc-chop” should be written with the lowercase letter “lnc-chop”. Done
Line 534: “endoplasmic reticulum stressed HCC cells” instead of “endoplasmic reticulum of stressed HCC cells”; Done
Line 602: “able to recruit RNase-H becouse brings …” instead of “able to recruit RNase-H enzyme because brings …”; Done
Line 638 – 641: Check spaces. Done
Font size problems
Line 431 “for example”;
Line 434 “(91)”;
Line 609 “lncRNA-NEAT1 and lncRNA-MALAT1”;
Line 631 “via”;
Line 635 “in vivo”;
Line 739 “of MIR155HG”
In the revised manuscript, all these font size issues have been addressed.
Reviewer 3 Report
Comprehensive and very well-written review.
Only to minor points:
- Page 1, line 33
cancer immunoediting (1). This dynamic and 34 non-linear process, that occurs during cancer onset, progression, and development of 35 drug resistance, includes three phases: …
It is only a concept, there are data published that argue against this concept. At least that should be stated.
- The font size of the text in the figures often too small.
Author Response
Reviewer #3
Comments and Suggestions for Authors
Comprehensive and very well-written review.
Only to minor points:
Page 1, line 33
cancer immunoediting (1). This dynamic and non-linear process, that occurs during cancer onset, progression, and development of drug resistance, includes three phases: …
It is only a concept, there are data published that argue against this concept. At least that should be stated.
We thanks the reviewer for this important point. We made clear that this is not unique mechanism of immunoresistance.
The font size of the text in the figures often too small.
Many thanks, wherever possible we enlarged the text in the figure
Round 2
Reviewer 2 Report
I consider the actual version of the manuscript suitable for publishing.